# Learning to (Learn at Test Time): RNNs with Expressive Hidden States

**Yu Sun** [* 1]  **Xinhao Li** [* 2]  **Karan Dalal** [* 3]  **Jiarui Xu** [2]  **Arjun Vikram** [1]  **Genghan Zhang** [1]  **Yann Dubois** [1]
**Xinlei Chen** [† 4]  **Xiaolong Wang** [† 2]  **Sanmi Koyejo** [† 1]  **Tatsunori Hashimoto** [† 1]  **Carlos Guestrin** [† 1]

## Abstract

Self-attention performs well in long context but has quadratic complexity. Existing RNN layers have linear complexity, but their performance in long context is limited by the expressive power of their hidden states. We present a practical framework for instantiating sequence modeling layers with linear complexity and expressive hidden states. The key idea is to make the hidden state a machine learning model itself, and the update rule a step of self-supervised learning. Since the hidden state is updated by training even on test sequences, our layers are called *Test-Time Training (TTT) layers*. We consider two instantiations: TTT-Linear and TTT-MLP, whose hidden state is a linear model and a two-layer MLP respectively. We evaluate our instantiations at the scale of 125M to 1.3B parameters, comparing with a strong Transformer and Mamba, a modern RNN. Similar to Transformer, TTT-Linear and TTT-MLP can keep reducing perplexity by conditioning on more tokens, while Mamba cannot after 16k context. TTT-MLP still faces challenges in memory I/O, but shows larger potential in long context, pointing to a promising direction for future research.

## 1. Introduction

This version of the paper has been abridged to fit the page limit of ICML camera ready. Please read our arXiv version instead: https://arxiv.org/abs/2407.04620.

In 2020, the OpenAI scaling law paper (Kaplan et. al (Kaplan et al., 2020)) showed that LSTMs (a type of RNN) could not scale similarly to Transformers or effectively use long context. Now, with modern RNNs and best practices,

we re-evaluate these findings in Figure 1.

On the left, we observe that Mamba (Gu & Dao, 2023) – one of the most popular RNNs today – scales similarly to a strong Transformer, showing great progress since the LSTMs in 2020. However, on the right, we observe the same issue with Mamba as Kaplan et al. did with LSTMs. Tokens later in a sequence should be easier to predict on average, since they condition on more information. This is indeed the case for Transformer, whose average perplexity at each token index decreases throughout its 32k context. In contrast, the same metric plateaus for Mamba after 16k.

This result represents an awkward reality for existing RNNs. On one hand, the main advantage of RNNs (vs. Transformers) is their linear (vs. quadratic) complexity. This asymptotic advantage is only realized in practice for long context, which according to Figure 8 is after 8k. On the other hand, once context is long enough, existing RNNs such as Mamba struggle to actually take advantage of the extra information being conditioned on.

The difficulty with long context is inherent to the very nature of RNN layers: Unlike self-attention, RNN layers have to compress context into a hidden state of fixed size. As a compression heuristic, the update rule needs to discover the underlying structures and relationships among thousands or potentially millions of tokens. This need is inherently challenging. In this paper, we begin with the observation that self-supervised learning can compress a massive training set into the weights of a model such as an LLM, which often exhibits deep understanding about the semantic connections among its training data – exactly what we need from a compression heuristic.

**TTT layers.** Motivated by this observation, we make the hidden state a machine learning model itself, and the update rule a step of self-supervised learning. Since the hidden state is updated by training even on test sequences, these RNN layers are called *Test-Time Training (TTT) layers*. We introduce two simple instantiations: TTT-Linear and TTT-MLP, where the hidden state is a linear model and a two-layer MLP, respectively. TTT layers can be integrated into any network architecture and optimized end-to-end, similar to RNNs layers and self-attention.

---

*: Core contributors. †: Joint advising.
[1]Stanford University [2]UC San Diego [3]UC Berkeley [4]Meta AI.

*Proceedings of the $42^{nd}$ International Conference on Machine Learning*, Vancouver, Canada. PMLR 267, 2025. Copyright 2025 by the author(s).

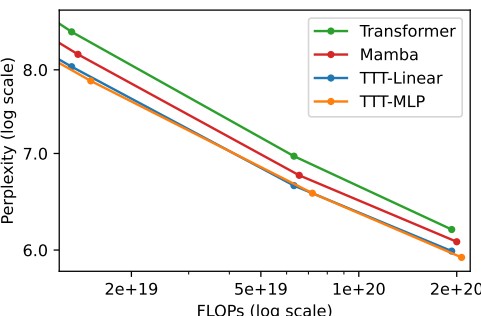 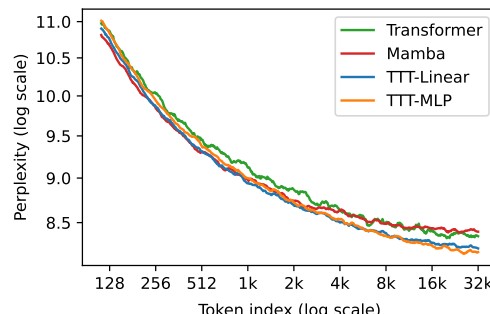

*Figure 1.* Comparing to Mamba, TTT-Linear and TTT-MLP have similar perplexity in 8k context (left) and better use of long context (right). Evaluations follow Kaplan et al. (Kaplan et al., 2020). **Left:** Scaling trends on the Pile with 8k context, zoomed in between 350M and 1.3B parameters. **Right:** Similar to Transformer, TTT-Linear and TTT-MLP can keep reducing perplexity by conditioning on more tokens, while Mamba cannot after 16k context. All methods have matched training FLOPs as Mamba 1.4B.

**Wall-clock time.** We apply two techniques to make TTT layers more efficient on modern GPUs and TPUs. First, similar to the standard practice of taking gradient steps on mini-batches of sequences during regular training for better parallelism, we use mini-batches of tokens during TTT. Second, we develop a dual form for operations inside each TTT mini-batch. The dual form is equivalent in output to the naive implementation, but trains more than $5\times$ faster on our TPUs.

**Contributions and limitations.** The idea of using linear models as hidden states has already been well studied in DeltaNet (Schlag et al., 2021; Yang et al., 2024). Since our first version was released, RNN layers with matrix (linear) hidden states have also been further advanced in Mamba 2 (Dao & Gu, 2024) and Gated DeltaNet (Yang et al., 2023). Compared to this line of work, our contribution is a practical framework that can instantiate arbitrary neural networks as hidden states. However, such instantiations can still require substantial wall-clock time, even after applying our improvements in efficiency. It remains to be seen whether our framework can produce instantiations that either overcome this limitation or offer benefits outweighing it.

## 2. Method

All sequence modeling layers can be viewed from the perspective of storing historic context into a hidden state, as shown in Figure 2. For example, RNN layers – such as LSTM (Hochreiter & Schmidhuber, 1997) and Mamba (Gu & Dao, 2023) layers – compress context into a state of fixed size across time. This compression has two consequences. On one hand, mapping an input token $x_t$ to output token $z_t$ is efficient, because both the update rule and output rule take constant time per token. On the other hand, the performance of RNN layers in long context is limited by the expressive power of its hidden state $s_t$.

Self-attention can also be viewed from the perspective above, except that its hidden state, commonly known as the Key-Value (KV) cache, is a list that grows linearly with $t$. Its update rule simply appends the current KV tuple to this list, and the output rule scans over all tuples up to $t$ to form the attention matrix. The hidden state explicitly stores all historic context without compression, making self-attention more expressive than RNN layers for long context. However, scanning this linearly growing hidden state also takes linearly growing time per token.

To remain both efficient and expressive in long context, we need a better compression heuristic. Specifically, we need to compress thousands or potentially millions of tokens into a hidden state that can effectively capture their underlying structures and relationships.

### 2.1. TTT as updating a hidden state

The process of parametric learning can be viewed as compressing a massive training set into the weights of a model. Specifically, we know that models trained with self-supervision can capture the underlying structures and relationships behind their training data (Le, 2013) – exactly what we need from a compression heuristic.

LLMs themselves are great examples. Trained with the self-supervised task of next-token prediction, their weights can be viewed as a compressed form of storage for existing knowledge on the internet. By querying LLMs, we can extract knowledge from their weights. More importantly, LLMs often exhibit a deep understanding of the semantic connections among existing knowledge to express new pieces of reasoning (Achiam et al., 2023).

Our key idea is to use self-supervised learning to compress the historic context $x_1, \ldots, x_t$ into a hidden state $s_t$, by making the context an unlabeled dataset and the state a model. Concretely, the hidden state $s_t$ is now equivalent to

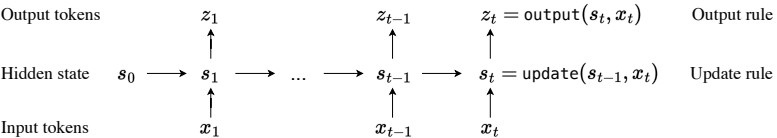

| | **Initial state** | **Update rule** | **Output rule** | **Cost** |
|---|---|---|---|---|
| **Naive RNN** | $s_0 = $ vector() | $s_t = \sigma\left(\theta_{ss} s_{t-1} + \theta_{sx} x_t\right)$ | $z_t = \theta_{zs} s_t + \theta_{zx} x_t$ | $O(1)$ |
| **Self-attention** | $s_0 = $ list() | $s_t = s_{t-1}$.append$(k_t, v_t)$ | $z_t = V_t \text{softmax}\left(K_t^T q_t\right)$ | $O(t)$ |
| **Naive TTT** | $W_0 = f$.params() | $W_t = W_{t-1} - \eta \nabla \ell(W_{t-1}; x_t)$ | $z_t = f(x_t; W_t)$ | $O(1)$ |

*Figure 2.* **Top**: A generic sequence modeling layer expressed as a hidden state that transitions according to an update rule. All sequence modeling layers can be viewed as different instantiations of three components in this figure: the initial state, update rule and output rule. **Bottom**: Examples of sequence modeling layers and their instantiations of the three components. Self-attention has a hidden state growing with context, therefore growing cost per token. Both the naive RNN and TTT layer compress the growing context into a hidden state of fixed size, therefore their cost per token stays constant.

$W_t$, the weights of a model $f$, which can be a linear model, a small neural network, or anything else. The output rule is simply: $z_t = f(x_t; W_t)$. Intuitively, the output token is just the prediction on $x_t$, made by $f$ with the updated weights $W_t$. The update rule is a step of gradient descent on some self-supervised loss $\ell$:

$$W_t = W_{t-1} - \eta \, \nabla \ell(W_{t-1}; x_t), \qquad (1)$$

with learning rate $\eta$.[1] From the compression point of view, every heuristic needs to decide which input to remember or forget. Our $W$ remembers inputs that produce large gradients – intuitively, inputs that make $W$ learn a lot.

One choice of $\ell$ is reconstructing $x_t$ itself. To make the learning problem nontrivial, we first process $x_t$ into a corrupted input $\tilde{x}_t$ (details in Subsection 2.3), then optimize:

$$\ell(W; x_t) = \|f(\tilde{x}_t; W) - x_t\|^2. \qquad (2)$$

Similar to denoising autoencoders (Vincent et al., 2008), $f$ needs to discover the correlations between dimensions of $x_t$ in order to reconstruct it from partial information $\tilde{x}_t$. We discuss more sophisticated formulations of the self-supervised task in Subsection 2.3.

As with other RNN layers and self-attention, our algorithm that maps an input sequence $x_1, \ldots, x_T$ to output sequence $z_1, \ldots, z_T$ can be programmed into the forward pass of a sequence modeling layer, using the hidden state, update rule, and output rule above. Even at test time, our new layer still trains a different sequence of weights $W_1, \ldots, W_T$ for every input sequence. Therefore, we call it the *Test-Time Training (TTT) layer*.

### 2.2. Training a network with TTT layers

The forward pass of a TTT layer also has a corresponding backward pass. Our forward pass only consists of standard differentiable operators except the gradient operator $\nabla$. However, $\nabla$ just maps one function to another, in this case $\ell$ to $\nabla \ell$, and $\nabla \ell$ is also composed of differentiable operators. Conceptually, calling backward on $\nabla \ell$ means taking gradients of gradients – a well explored technique in meta-learning (Maclaurin et al., 2015).

TTT layers have the same interface as RNN layers and self-attention, therefore can be replaced in any larger network architecture, which usually contains many of these sequence modeling layers. Training a network with TTT layers also works the same way as training any other language model, such as a Transformer. The same data, recipe, and objective such as next-token prediction can be used to optimize parameters of the rest of the network.

We refer to training the larger network as the *outer loop*, and training $W$ within each TTT layer as the *inner loop*. An important difference between the two nested learning problems is that the inner-loop gradient $\nabla \ell$ is taken w.r.t. $W$, the parameters of $f$, while the outer-loop gradient is taken w.r.t the parameters of the rest of the network, which we will denote by $\theta_{\text{rest}}$. Throughout this paper, outer-loop parameters are always denoted by $\theta$ with various subscripts.

### 2.3. Learning a self-supervised task for TTT

Arguably the most important part of TTT is the self-supervised task, because it determines the kind of features that $W$ will learn from the test sequence. So how should we design this task? The final goal of TTT is for $z_t = f(x_t; W_t)$ to perform well on language modeling. Instead of handcrafting a self-supervised task from human priors, we take a more end-to-end approach – directly optimizing the self-supervised task for the final goal of next-token prediction.

---

[1] For now, consider $W_0 = 0$. We will discuss more sophisticated techniques for initializing $W$ in Subsection 2.7.

Concretely, we learn the self-supervised task as part of the outer loop. Starting from the naive reconstruction task in Equation 2, we add some outer-loop parameters to make this task learnable. In Subsection 2.1, we did not specify the corruption that produces $\tilde{x}_t$ from $x_t$. One design is to make it a low-rank projection $\tilde{x}_t = \theta_K x_t$, where $\theta_K$ is a learnable matrix.[2] Following the terminology of multi-view reconstruction, $\theta_K x_t$ is called a *training view*.

Moreover, perhaps not all the information in $x_t$ is worth remembering, so the reconstruction label can be another low-rank projection $\theta_V x_t$ instead of $x_t$. Here $\theta_V x_t$ is called the *label view*, where $\theta_V$ is also learnable. In summary, our new self-supervised loss is:

$$\ell(W; x_t) = \left\| f\left(\theta_K x_t; W\right) - \theta_V x_t \right\|^2. \qquad (3)$$

Since both $W$ and various $\theta$s appear together in Equation 3, we emphasize again their difference in nature. In the inner loop, only $W$ is optimized, therefore written as an argument of $\ell$; the $\theta$s are "hyper-parameters" of this loss function. In the outer loop, $\theta_K, \theta_V, \theta_Q$ are optimized alongside $\theta_{\text{rest}}$, and $W$ is merely a hidden state, not a parameter.

Lastly, the training view $\theta_K x_t$ has fewer dimensions than $x_t$, so we can no longer use the output rule in Subsection 2.1. The simplest solution is to create a *test view* $\theta_Q x_t$, and change our output rule to:

$$z_t = f\left(\theta_Q x_t; W_t\right). \qquad (4)$$

This solution has an additional benefit. The training and label views specify the information in $x_t$ that is compressed into $W_t$ and propagated forward through time. The test view specifies potentially different information that is mapped to the current output token $z_t$ and propagated forward through network layers, therefore adds more flexibility to the self-supervised task.

### 2.4. Parallelization with mini-batch TTT

The naive TTT layer developed so far is already efficient in the number of floating point operations (FLOPs). However, its update rule $W_t = W_{t-1} - \eta \nabla l(W_{t-1}; x_t)$ cannot be parallelized, because $W_t$ depends on $W_{t-1}$ in two places: before the minus sign and inside $\nabla l$. Since $\nabla l$ contains the bulk of the computation, we focus on making this second part parallel.

We approach this systems challenge through concepts in the TTT framework. There are many variants of gradient descent (GD). Its general update rule can be expressed as:

$$W_t = W_{t-1} - \eta\, G_t = W_0 - \eta \sum_{s=1}^{t} G_s, \qquad (5)$$

[2] The subscript $K$ hints at a connection to self-attention, as we will establish in Subsection 2.6.

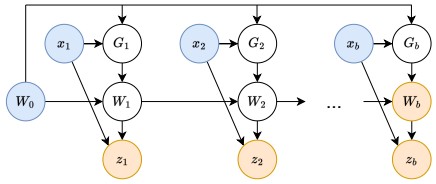

*Figure 3.* High-level computation graph of the first TTT mini-batch, where nodes are variables and edges are computations. The blue nodes are input variables, and yellow are output.

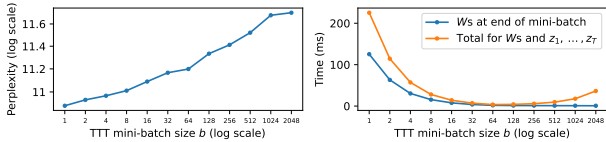

*Figure 4.* Ablations on TTT mini-batch size $b$, where $b = 1$ is online GD and $b = T$ is batch GD. We choose $b = 16$ for all experiments in this paper. **Left**: Smaller $b$ improves perplexity since more GD steps are taken. The perplexity of 11.09 at $b = 16$ corresponds to the final result of TTT-Linear in Figure 6. **Right**: Forward time in dual form, with context length $T = 2048$. Total time (orange) can be decomposed into time for computing the $W$s at the end of every mini-batch (blue) and time for $z_1, \ldots, z_T$.

where $G_t$ is the descent direction. Note that once we have calculated $G_t$ for $t = 1, \ldots, T$, we can then obtain all the $W_t$s through a cumsum by the second half of Equation 5. Our naive update rule, known as *online gradient descent*, uses $G_t = \nabla l(W_{t-1}; x_t)$.

To parallelize $G_t$ for $t = 1, \ldots, T$, we can take all of them w.r.t. $W_0$. This variant with $G_t = \nabla \ell(W_0; x_t)$ is known as *batch gradient descent*, since $\sum_{s=1}^{t} \nabla \ell(W_0; x_s)$ is the same as the gradient w.r.t. $W_0$ over $x_1, \ldots, x_t$ as a batch. However, in batch GD, $W_t$ is effectively only one gradient step away from $W_0$, in contrast to online GD, where $W_t$ is $t$ steps away from $W_0$. Therefore, batch GD has a smaller effective search space, which ends up hurting performance for language modeling.

Our proposed solution – *mini-batch gradient descent* – is shown in Figure 3. Denote the TTT batch size by $b$. We use $G_t = \nabla \ell(W_{t'}; x_t)$, where $t' = t - \mod(t, b)$ is the last timestep of the previous mini-batch (or 0 for the first mini-batch), so we can parallelize $b$ gradient computations at a time. Empirically, $b$ controls a trade-off between speed and quality, as shown in Figure 4. We chose $b = 16$ for all experiments in this paper.

### 2.5. Dual form

The parallelization introduced above is necessary but not sufficient for efficiency in wall-clock time. Modern accelerators specialize in matrix-matrix multiplications, known as

matmuls. For example, the NVIDIA A100 GPU contains highly optimized units called TensorCores that can only perform a single operation – multiplying two matrices each of size $16 \times 16$. Without enough of these matmuls, the TensorCores are idle, and most of the potential for the A100 is unrealized.

Unfortunately, the TTT layer developed so far even with mini-batch still has very few matmuls. Consider the simplest case of $\ell$, where $\theta_K = \theta_V = \theta_Q = I$, for only the first TTT mini-batch of size $b$. In addition, consider $f$ as a linear model. Copying Equation 2, our loss at time $t$ is: $\ell(W_0; x_t) = \|W_0 x_t - x_t\|^2$. As discussed in Subsection 2.4, we can parallelize the computation of: $G_t = 2(W_0 x_t - x_t)x_t^T$, for $t = 1, \dots, b$. However, we cannot compute all $b$ of the $G_t$s through a single matmul. Instead, we need $b$ outer products to compute them one by one. To make matters worse, for each $x_t \in \mathbb{R}^d$, $G_t$ is $d \times d$, which incurs much heavier memory footprint and I/O cost than $x_t$ for large $d$.

To solve these two problems, we make a simple observation: We do not actually need to materialize $G_1, \dots, G_b$ as long as we can compute $W_b$ at the end of the mini-batch, and the output tokens $z_1, \dots, z_b$ (see Figure 3). Now we demonstrate these computations with the simplified TTT-Linear case above. Denote $X = [x_1, \dots, x_b]$, then: $W_b = W_0 - 2\eta(W_0 X - X)X^T$. So $W_b$ can be conveniently computed with a matmul. To compute $Z = [z_1, \dots, z_b]$, we know that:

$$z_t = f(x_t; W_t) = W_0 x_t - 2\eta \sum_{s=1}^{t}(W_0 x_s - x_s)x_s^T x_t. \quad (6)$$

Denote $\delta_t = \sum_{s=1}^{t}(W_0 x_s - x_s)x_s^T x_t$ and the matrix $\Delta = [\delta_1, \dots, \delta_b]$. We can derive that:

$$\Delta = (W_0 X - X)\,\mathrm{mask}\left(X^T X\right), \quad (7)$$

where mask is the upper triangular mask with zeros (similar to the attention mask, but with zeros instead of infinities), and the term $W_0 X - X$ can be reused from the computation of $W_b$. Now $\Delta$ is also conveniently computed with matmuls. Plugging $\Delta$ back into Equation 6, we obtain $Z = W_0 X - 2\eta\Delta$.

We call this procedure the *dual form*, in contrast to the *primal form* before this subsection, where the $G$s and $W$s are explicitly materialized. As discussed, the two forms are equivalent in output. The terminology of primal and dual follows prior work that has explored similar mathematical formulations outside of TTT (Irie et al., 2022; Bishop & Nasrabadi, 2006; Rosenblatt, 1958). In Appendix A, we show that the dual form still works when $f$ is a neural network with nonlinear layers.

Time complexity of the primal form within a TTT mini-batch is $O(b \times d^2)$. Time complexity of the dual form

is $O(b \times d^2)$ for computing $W_b$ alone, then an additional $O(b^2 \times d)$ for computing $z_1, \dots, z_b$. Compared to the primal, the dual form sacrifices theoretical complexity for hardware utilization. In practice, $d$ is typically a few hundred and $b$ is chosen to be only 16. As a consequence, wall-clock time for computing $z_1, \dots, z_b$ is relatively small, as observed in the right panel of Figure 4. In our JAX implementation, training with the dual form is more than $5\times$ faster than with primal.

## 2.6. Theoretical equivalences

In Subsection 2.1, we mentioned that $f$ can be a linear model or a neural network. In Subsection 2.4, we also discussed three variants of the update rule: online GD, batch GD, and mini-batch GD. Each of these $2 \times 3$ combinations induces a different instantiation of the TTT layer. We now show that among these induced instantiations, the TTT layer with a linear model and batch GD is equivalent to linear attention (Katharopoulos et al., 2020).

**Theorem 1.** *Consider the TTT layer with $f(x) = Wx$ as the inner-loop model, batch gradient descent with $\eta = 1/2$ as the update rule, and $W_0 = 0$. Then, given the same input sequence $x_1, \dots, x_T$, the output rule defined in Equation 4 produces the same output sequence $z_1, \dots, z_T$ as linear attention.*

*Proof.* By definition of $\ell$ in Equation 3, $\nabla\ell(W_0; x_t) = -2(\theta_V x_t)(\theta_K x_t)^T$. By definition of batch GD: $W_t = \sum_{s=1}^{t}(\theta_V x_s)(\theta_K x_s)^T$. Plugging $W_t$ into the output rule in Equation 4, we obtain the output token: $z_t = f(\theta_Q x_t; W_t) = \sum_{s=1}^{t}(\theta_V x_s)(\theta_K x_s)^T(\theta_Q x_t)$, which is the definition of linear attention. $\square$

In Table 1, we first empirically verify the equivalence above with an improved implementation of linear attention. Then, to illustrate the contribution of each of our components (including some that will be introduced in the next subsection), we add them row by row to the TTT layer that is equivalent to linear attention, and ultimately obtain our proposed instantiation called *TTT-Linear*. The change from batch GD to mini-batch GD contributes the most improvement by a large margin.

While the space of models $\times$ optimizers is already large, machine learning is much richer than optimizing the parameters $W_t$ of a model $f$. There are also nonparametric learners, such as nearest neighbors, support vector machines (SVMs), and kernel ridge regression. By definition, nonparametric learners do not have parameters $W_t$, and instead directly uses training data $x_1, \dots, x_t$. Hence we use the notation $f(x; x_1, \dots, x_t)$. We now show that for a particular nonparametric learner, the induced TTT layer is equivalent to self-attention.

**Theorem 2.** *Consider the TTT layer with the Nadaraya-*

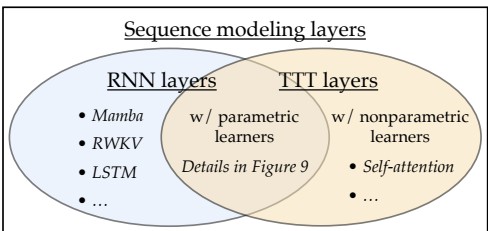

*Figure 5.* RNN layers and TTT layers are both subsets of sequence modeling layers. RNN layers have a hidden state that is fixed in size across time. TTT layers with parametric learners are also RNN layers, since their hidden state is also fixed in size. TTT layers with nonparametric learners can represent self-attention, as discussed in Subsection 2.6.

*Watson estimator (Bierens, 1988; Cai, 2001), defined as:*

$$f(x; x_1, \ldots, x_t) = \frac{1}{\sum_{s=1}^{t} \kappa(x, x_s)} \sum_{s=1}^{t} \kappa(x, x_s)\, y_s, \quad (8)$$

*where $y_s = \theta_V x_s$, and*

$$\kappa\left(x, x'; \theta_K, \theta_Q\right) \propto e^{(\theta_K x)^T \theta_Q x'} \quad (9)$$

*is a kernel with bandwidth hyper-parameters $\theta_K$ and $\theta_Q$. Then given the same input sequence $x_1, \ldots, x_T$, the output rule defined in Equation 4 produces the same output sequence $z_1, \ldots, z_T$ as self-attention.*

*Proof.* Plugging $y_s$ and $\kappa$ above into Equation 8 gives us the definition of self-attention. ☐

Appendix B contains a detailed explanation of the Nadaraya-Watson estimator and kernel $\kappa$ above. In contrast to Theorem 1, Theorem 2 does not produce a different implementation from attention.

### 2.7. Implementation details

**Instantiations of $f$.** We propose two variants of TTT layers – TTT-Linear and TTT-MLP, differing only in their instantiations of $f$. For TTT-Linear, $f_{\texttt{lin}}(x) = Wx$, where $W$ is square. For TTT-MLP, $f_{\texttt{MLP}}$ has two layers similar to the MLPs in Transformers. Specifically, the hidden dimension is $4\times$ the input dimension, followed by a GELU activation (Hendrycks & Gimpel, 2016). For better stability during TTT, $f$ always contains a Layer Normalization (LN) and residual connection. That is, $f(x) = x + \text{LN}(f_{\texttt{res}}(x))$, where $f_{\texttt{res}}$ can be $f_{\texttt{lin}}$ or $f_{\texttt{MLP}}$.

**Learnable $W_0$.** The TTT initialization $W_0$ is shared between all sequences, even though subsequent weights $W_1, \ldots, W_T$ are different for each input sequence. Instead of setting $W_0 = 0$, we can learn it as part of the outer loop. Since outer-loop parameters are always denoted by $\theta$s instead of $W$s, we assign an alias $\theta_{\text{init}} = W_0$. In practice, $\theta_{\text{init}}$

| Configuration | Ppl. | Diff. |
|---|---|---|
| Linear attention (Katharopoulos et al., 2020) | 15.91 | - |
| Linear attn. improved | 15.23 | −0.68 |
| TTT equivalence | 15.23 | 0 |
| + learnable $W_0$ | 15.27 | +0.04 |
| + LN and residual in $f$ | 14.05 | −1.22 |
| + mini-batch TTT | 12.35 | −1.70 |
| + learnable $\eta$ | 11.99 | −0.36 |
| + Mamba backbone | 11.09 | −0.90 |

*Table 1.* Ablations on improving from linear attention. All models here have 125M parameters, and are trained according to the recipe in Subsection 3.1. The last row, with perplexity 11.09, is the final result of TTT-Linear in Figure 6. Starting from the equivalence discussed in Subsection 2.6, learnable $W_0$ hurts slightly, but the rows below cannot train stably without it. The biggest improvement comes from mini-batch TTT (changing from $b = T = 2048$ to $b = 16$). The second comes from instantiating the inner model $f$ with LN and residual connection.

adds a negligible amount of parameters comparing to the reconstruction views $\theta_K, \theta_Q, \theta_V$, because both its input and output are low dimensional. Empirically, we observe that learning $W_0$ significantly improves training stability.

**Learnable $\eta$.** The learning rate is usually the most important hyper-parameter for gradient descent, so we experiment with learning the inner-loop learning rate $\eta$ in Equation 5 as part of the outer loop. We make $\eta$ a function of the input token (therefore different across time) for additional flexibility. Concretely, we design $\eta(x) = \eta_{\text{base}}\, \sigma(\theta_{\text{lr}} \cdot x)$, where the learnable vector $\theta_{\text{lr}}$ is an outer-loop parameter, $\sigma$ is the sigmoid function, and the scalar $\eta_{\text{base}}$ is the base learning rate, set to 1 for TTT-Linear and 0.1 for TTT-MLP. Alternatively, $\eta(x)$ can also be interpreted as a gate for $\nabla \ell$.

**Backbone architecture.** The cleanest way to integrate any RNN layer into a larger architecture would be to directly replace self-attention in a Transformer, known in this context as a backbone. However, existing RNNs such as Mamba (Gu & Dao, 2023) and Griffin (De et al., 2024) all use a different backbone from Transformers. Most notably, their backbone contains temporal convolutions before the RNN layers, which might help collect local information across time. After experimenting with the Mamba backbone, we find that it also improves perplexity for TTT layers, so we incorporate it into our proposed method. See Figure 9 (in Appendix) for details.

## 3. Experiments

We evaluate TTT-Linear and TTT-MLP by comparing with two baselines – Transformer and Mamba, a modern RNN. Our main codebase is based on EasyLM (Geng, 2023), an open-source project for training and serving LLMs in JAX.

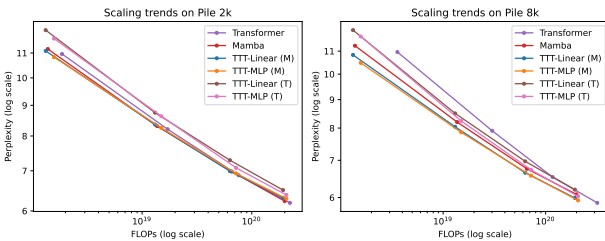 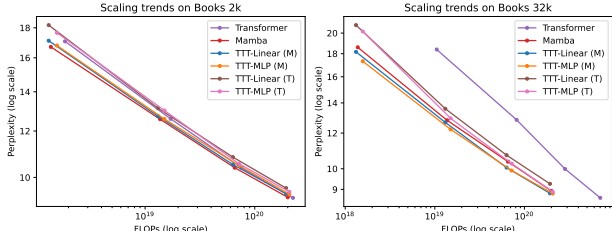

*Figure 6.* Evaluations for context lengths 2k and 8k on the Pile. Details in Subsection 3.1. TTT-Linear has comparable performance as Mamba at 2k context, and better performance at 8k.

*Figure 7.* Evaluations for context lengths 2k and 32k on Books. Details in Subsection 3.2. Our complete results for context lengths 1k, 2k, 4k, 8k, 16k, 32k, including Transformer finetuning, are in Figure 11 (in Appendix).

**Datasets.** Following the Mamba paper (Gu & Dao, 2023), we perform standard experiments with 2k and 8k context lengths on the Pile (Gao et al., 2020), a popular dataset of documents for training open-source LLMs (Black et al., 2022). However, the Pile contains few sequences of length greater than 8k (de Vries, 2023). To evaluate capabilities in long context, we also experiment with context lengths ranging from 1k to 32k in $2\times$ increments, on a subset of the Pile called Books3, which has been widely used to train LLMs in long context (Liu et al., 2024).

**Backbone architecture.** As discussed in Subsection 2.7, Transformer and Mamba use different backbones, and TTT-Linear and TTT-MLP always use the Mamba backbone unless noted otherwise. As an ablation study, Figure 6 and Figure 7 contain TTT layers within the Transformer backbone. When a figure contains both the Transformer backbone and Mamba backbone, we denote them by *(T)* and *(M)*, respectively.

**Protocols.** To ensure fairness to our baselines, we strictly follow the evaluation protocols in the Mamba paper when possible. For each evaluation setting (e.g., dataset, context length, and method), we experiment with four model sizes: 125M, 350M, 760M, and 1.3B parameters. For Mamba, the corresponding sizes are 130M, 370M, 790M, and 1.4B, as Mamba does not follow the Transformer configurations. All models are trained with the Chinchilla recipe described in the Mamba paper and reproduced in our Appendix C.

### 3.1. Short context: the Pile

From Figure 6, we make a few observations:

- At 2k context, TTT-Linear (M), Mamba, and Transformer have comparable performance, as the lines mostly overlap. TTT-MLP (M) performs slightly worse under large FLOP budgets. Even though TTT-MLP has better perplexity than TTT-Linear at every model size, the extra cost in FLOPs offsets the advantage.

- At 8k context, both TTT-Linear (M) and TTT-MLP (M) perform significantly better than Mamba, in contrast to the observation at 2k. Even TTT-MLP (T) with the Trans-

former backbone performs slightly better than Mamba around 1.3B. A robust phenomenon we observe throughout this paper is that as context length grows longer, the advantage of TTT layers over Mamba widens.

- At 8k context, Transformer still has good (if not the best) perplexity at every model size, but its line is not competitive because of the cost in FLOPs.

**Effect of backbone.** Switching the TTT layers from Mamba backbone into Transformer backbone has two effects. First, TTT layers with Mamba backbone perform better in our evaluations so far. Second, with Mamba backbone, TTT-MLP at best is only comparable to TTT-Linear; but with Transformer backbone, TTT-MLP is clearly better. We hypothesize that the temporal convolutions in the Mamba backbone help more when the sequence modeling layer has a less expressive hidden state. The linear model is less expressive than the MLP, therefore benefits more from the convolutions. We will revisit this hypothesis in the next subsection.

### 3.2. Long context: Books

To evaluate capabilities in long context, we experiment with context lengths ranging from 1k to 32k in $2\times$ increments, using a popular subset of the Pile called Books3. The training recipe here is the same as that for Pile. From the subset of results in Figure 7, we make a few observations:

- At 2k context on Books, all the observations from Pile 2k still hold, except that Mamba now performs slightly better than TTT-Linear (whereas their lines roughly overlapped for Pile 2k).

- At 32k context, both TTT-Linear (M) and TTT-MLP (M) perform better than Mamba, similar to the observation from Pile 8k. Even TTT-MLP (T) with the Transformer backbone performs slightly better than Mamba at 32k context.

- TTT-MLP (T) is only slightly worse than TTT-MLP (M) at 1.3B scale. As discussed, it is hard to derive an empirical scaling law due to the lack of a clean linear fit.

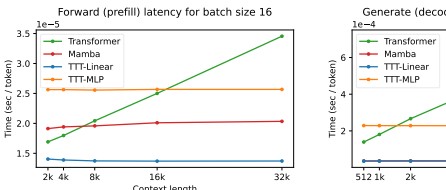

*Figure 8.* Latency on an NVIDIA A100 GPU with 80G HBM and PCIe connections.

However, the strong trend for TTT-MLP (T) suggests that the Transformer backbone might be more suitable for larger models and longer context beyond our evaluations.

We only ablate the backbones for 2k and 32k due to the cost of training LLMs. For future work, we believe that given TTT layers with even more expressive hidden states, the Mamba backbone with convolutions will be unnecessary.

**Transformer finetuning.** While we have been training Transformers from scratch following the Mamba paper, in practice this approach is rarely used for long context. The standard practice is to train a Transformer in short context, then finetune in long context. To reflect this practice, we add another baseline, *TF finetune*, for context lengths 4k and above. This baseline starts from the model trained (according to the Chinchilla recipe) on Books 2k, then uses 20% more tokens to finetune at the designated context length, following the Llama Long paper (Xiong et al., 2023). See details of the TF finetune recipe in Appendix C.

**Experiments in Figure 1 (right).** Compared to TTT-Linear, TTT-MLP with matched FLOPs performs worse at short context but better at long context. This observation matches our expectation that the MLP as hidden state is more expressive than the linear model: The larger capacity of a more expressive hidden state is well-utilized in long context (therefore an advantage), but redundant in short context (therefore a disadvantage in our setting with matched FLOPs). The Transformer in this figure is TF finetune, which is the stronger baseline in 32k context. Details of the experiments in Figure 1 are included in Appendix C. Our complete results for context lengths 1k, 2k, 4k, 8k, 16k, 32k, including TF finetune, are in Figure 11 (in Appendix).

### 3.3. Wall-clock time

LLM training and inference can be decomposed into forward, backward, and generate. Prompt processing during inference, also known as prefill, is the same operation as forward during training, except that the intermediate activations do not need to be stored for backward. Since both forward (during training and inference) and backward can be parallelized, we use the dual form. Generating new tokens, also known as decode, is inherently sequential, so we use the primal form.

Due to resource constraints, our experiments are written in JAX and run on TPUs. On a v5e-256 TPU pod, the Transformer baseline takes 0.30s per iteration of training at context 2k, while TTT-Linear takes 0.27s per iteration, already 10% faster without any systems optimization. However, Mamba (implemented in PyTorch, Triton, and CUDA) can only run on GPUs, so for fair comparison, we also rewrite our method into GPU kernels. We only write inference kernels for this work because the training kernel would require substantial effort and cannot be used on our TPUs.

Figure 8 shows the latency of our inference kernel for forward (prefill) and generate (decode). All models are 1.3B (1.4B for Mamba). As expected, time per token grows linearly for Transformer as the context length increases, but stays roughly constant for the other methods. Note that our Transformer baseline is significantly faster that in the Mamba paper, because we use vLLM (Kwon et al., 2023), a state-of-the-art serving system, instead of the HuggingFace Transformer (Wolf et al., 2019).

## 4. Related Work

### 4.1. Learning at Test Time

The idea of learning at test time has a long history in machine learning. One of the earliest versions of this idea is called local learning (Bottou and Vapnik (Bottou & Vapnik, 1992)): For each test input, train on its neighbors before making a prediction. This procedure has been effectively applied to models ranging from SVMs (Zhang et al., 2006) to modern LLMs (Hardt & Sun, 2023). Next, we discuss two relevant lines of work in detail: test-time training and fast weights.

#### 4.1.1. TEST-TIME TRAINING

The core idea of *Test-Time Training* (TTT) is that each test instance defines its own learning problem, where this test instance alone is the target of generalization (Sun et al., 2020). Concretely, for each test instance $x$, the conventional practice is to predict $f(x)$, using a predictor $f$ that is optimized for all training instances on average. TTT first formulates a learning problem defined by $x$, then trains a model $f_x$ on $x$ (often with $f$ as initialization), and predicts $f_x(x)$.

Since the test instance comes without its label, the learning problem can only be formulated with a self-supervised task. Prior work has shown that TTT with reconstruction significantly improves performance especially on outliers (Gandelsman et al., 2022). Improvements become even more pronounced when testing on video frames that arrive in a stream and TTT is autoregressive (Wang et al., 2023), as $f_t$ is trained on past frames $x_1, \ldots, x_t$. The autoregressive connection makes (Wang et al., 2023) most relevant to our paper. Conceptually, the biggest difference between our pa-

per and prior work is that our reconstruction task is learned in an outer loop, instead of handcrafted with human priors.

### 4.1.2. FAST WEIGHTS

The general idea of *fast weights* is to update the parameters of a "fast" model on only the most relevant data, as opposed to the conventional practice of updating a "slow" model on all data (Tieleman & Hinton, 2009). This idea has existed since the 1980s (Hinton & Plaut, 1987). The most relevant data can be the test instance itself, therefore TTT can be viewed as a special case of fast weights. Compared to fast weights, TTT embraces the idea of formulating an explicit learning problem, where the test instance is the target of generalization. Our update rule is also an explicit step of optimization.

The idea of *fast weight programmers* (FWPs) is to update the fast weights with a "slow" model (Schmidhuber, 1992). As a modern example for language modeling, Clark et al. (Clark et al., 2022) give a Transformer a final layer of fast weights, whose initialization is trained as slow weights. Our inner-loop weights $W$ can be viewed as "fast" and outer-loop weights $\theta$ as "slow". Therefore, networks containing TTT layers can be viewed as a special case of FWPs (Kirsch & Schmidhuber, 2021), similar to how TTT can be viewed as a special case of fast weights.

Modern RNN layers such as linear attention (Katharopoulos et al., 2020; Schlag et al., 2020) and DeltaNet (Schlag et al., 2021; Yang et al., 2024) are inspired by the idea of FWPs. Given their relevance to our work, we discuss these modern RNN layers in detail in the next subsection.

### 4.2. Modern RNN layers

Our baseline, Mamba (Gu & Dao, 2023), is only one of the many recent RNN layers that inherit the linear (matrix) hidden states of linear attention (Katharopoulos et al., 2020; Schlag et al., 2020). Some more recent examples are RWKV (Peng et al., 2024), xLSTM (Beck et al., 2024), and Gated Linear Attention (GLA) (Yang et al., 2023). The most relevant work is DeltaNet (Schlag et al., 2021), which is equivalent to TTT-Linear with inner-loop mini-batch size 1, without the Layer Norm and residual connection. (Yang et al., 2024) further improve the performance of DeltaNet and enable parallelized updates across tokens (in our terms, across inner loop mini-batches). Since our first version was released, RNN layers with matrix (linear) hidden states have also been further advanced in Mamba 2 (Dao & Gu, 2024) and Gated DeltaNet (Yang et al., 2023).

Compared to this line of work, our contribution is a practical framework that can instantiate arbitrary neural networks as hidden states. However, such instantiations can still require substantial wall-clock time, even after applying our

improvements in efficiency. For example, TTT-MLP is effective in terms of FLOPs, as shown in Figure 1. But the additional complexity of the MLP structure increases wall-clock time much more relative to FLOPs, as shown in Figure 8. It remains to be seen whether our framework can produce instantiations that either overcome this limitation or offer benefits outweighing it.

### 4.3. Learning to Learn

For decades, researchers have been arguing that learning to learn, also known as meta-learning or bi-level optimization, should be a critical component of intelligence (Schmidhuber, 1987; Bengio et al., 1990; Thrun & Pratt, 1998; Lake et al., 2017). In prior work such as (Andrychowicz et al., 2016), (Finn et al., 2017) and (Metz et al., 2018), the inner loop learns from an entire dataset at a time instead of a sequence, so the outer loop needs a collection of datasets or tasks. In short, the outer loop is "one level above" regular training. Since it is hard to collect millions of datasets, this outer loop is hard to scale.

In contrast, for TTT, each sequence itself is a dataset and defines its own generalization problem. The inner loop is "one level below" regular training, so our outer loop is only another solution to the canonical problem of supervised learning, instead of a new problem setting like generalization across datasets.

## 5. Future work

The search space for effective instantiations inside this framework is huge, and our paper has only taken a baby step. Fortunately, if our perspective holds, then heuristics from regular training can transfer to test-time training, and search can be efficient. Next we outline some especially promising directions for future work:

- **Systems optimization.** Our systems optimization in Subsection 3.3 has been preliminary at best, and there are many ways to improve it. In addition, pipeline parallelism through time might allow us to process long sequences of millions of tokens on multiple devices together.

- **Longer context and larger models.** Constrained by our academic resources, we have not trained with millions or billions in context length, which would also require larger models according to Figure 12. The advantage of TTT layers should become more pronounced in longer context.

- **More ambitious instantiations of $f$.** When context length becomes longer, $f$ would also need to be larger. For video tasks and embodied agents, whose context length can easily scale up to millions or billions, $f$ could be a convolutional neural network.

## Impact statement

This paper presents work whose goal is to advance the field of Machine Learning. There are many potential societal consequences of our work, none which we feel must be specifically highlighted here.

## Acknowledgements

Part of the compute for this project is generously supported by the Google TPU Research Cloud program and Hyperbolic Labs. XW is supported, in part, by the Amazon Research Award, the Cisco Faculty Award and the Qualcomm Innovation Fellowship. SK acknowledges support by NSF 2046795 and 2205329, NIFA award 2020-67021-32799, the Alfred P. Sloan Foundation, and Google Inc. TH is supported by a Sony Faculty Innovation Award and a gift from Panasonic. CG acknowledges support by the Air Force Office of Scientific Research (AFOSR), FA9550-20-1-0427, Stanford Human-Centered Artificial Intelligence (HAI) Institute, and gifts from Google and IBM.

We would like to thank Rohan Taori, Xuechen Li, Allan Zhou, Ke Chen, and Guandao Yang for many helpful discussions, Menghao Guo for help with code release, Xinyang Geng for help with EasyLM, Hao Liu for help with the LWM codebase, David Hall for help with Levanter, Yossi Gandelsman and Yutong Bai for help at an early stage of the project, Mert Yuksekgonul for help with figures in the paper, Horace He, Ben Spector, and Azalia Mirhoseini for help with systems, Sharad Vikram and Roy Frostig for answering our questions about JAX and Pallas, Albert Gu and Tri Dao for helping us reproduce experiments in the Mamba paper, and Kilian Weinberger and Percy Liang for advice on presentation. Yu Sun is grateful to his PhD advisors, Alexei A. Efros and Moritz Hardt, for their many insights from years ago that eventually became part of this paper.

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

# Appendix

## A. Dual Form

Here we derive the dual form for general MLPs of arbitrary depth, with nonlinear activations.

Without loss of generality, consider $\eta = 1$ for convenience, and consider only the first mini-batch, where $t = 1, \ldots, b$. Denote:

$$\hat{x}_t = \theta_K x_t, \quad y_t = \theta_V x_t, \quad \bar{x}_t = \theta_Q x_t.$$

Also denote $\hat{X} = [\hat{x}_1, \ldots, \hat{x}_b]$, and $Y$ and $\bar{X}$ analogously. In general, uppercase letters denote matrices whose columns are vectors denoted by the corresponding lowercase letter.

For a network with $K$ layers, denote the initial parameters in layer $k$ by $W_0^k$. Our convention is to use superscripts for the layer and subscripts for time.

### A.1. Forward pass

During the initial forward pass of TTT, we denote the input to layer $k$ by $\hat{X}^k = [\hat{x}_1^k, \ldots, \hat{x}_b^k]$, with $\hat{X}^1 = \hat{X}$. Now we write the forward pass of TTT using these notations.

For $k = 1, \ldots, K$:

- $Z^k = W_0^k \hat{X}^k$
- $\hat{X}^{k+1} = \sigma_k \left( Z^k \right)$

where $\sigma_k$ for $k = 1, \ldots, K$ can be any element-wise operation ($\mathbb{R} \mapsto \mathbb{R}$) with derivative $\sigma'$.

Given $\hat{X}^{K+1}$, we compute the loss:

$$l = \frac{1}{2} \ell \left( W_0^1, \ldots, W_0^K; X \right) = \frac{1}{2} \left\| \hat{X}^{K+1} - Y \right\|_F^2 = \sum_{t=1}^{b} l_t,$$

where $l_t = \frac{1}{2} \|\hat{x}_t^K - y_t\|^2$ is the same as defined in Equation 3, except scaled by $1/2$ for convenience.

All the operations above (except $\sigma$) are `matmuls` and `sums`, therefore are hardware efficient. Both the primal form and the dual form share these initial operations.

### A.2. Primal form

The primal form first computes $G_t^k = \nabla_{W_0^k} l_t$ for $t = 1, \ldots, b$, then updates $W_t^k = W_0^k - \sum_{s=1}^{t} G_s^k$. Finally, given $\bar{X}^1 = [\bar{x}_1^1, \ldots, \bar{x}_b^1] = \bar{X}$, the primal form repeats the forward pass with the updated $W$s.

For $k = 1, \ldots, K$:

- $\bar{z}_t^k = W_t^k \bar{x}_t^k$, for $t = 1, \ldots, T$
- $\bar{x}_t^{k+1} = \sigma_k(\bar{z}_t^k)$, for $t = 1, \ldots, T$

where $\bar{X}^{K+1} = [\bar{x}_1^{k+1}, \ldots, \bar{x}_b^{k+1}]$ contains the output tokens.

Note that a standard backward pass only computes the sum of the gradients:

$$\nabla_{W_0^k} l = \sum_{t=1}^{b} \nabla_{W_0^k} l_t = \sum_{t=1}^{b} G_t^k,$$

so the computation of the individual terms in the sum $G_t^k$ for $t = 1, \ldots, b$ cannot be batched together into `matmuls`. Similarly, the forward pass in primal form uses a different $W_t$ for each $\bar{x}_t$, therefore also cannot be batched in the same way as a standard forward pass. These non-standard passes have poor hardware efficiency.

## A.3. Dual form

As discussed in Subsection 2.5, the goal of the dual form is to compute $\bar{X}^{K+1}$ and $W_b^1, \ldots, W_b^K$ with only `matmuls` and light-weight operations such as `sums`, $\sigma$, and $\sigma'$. To achieve this goal, we avoid explicitly computing the intermediate variables: $G_t^k$ and $W_t^k$ for $t = 1, \ldots, b$.

The dual form first computes $\nabla_{\hat{X}^{K+1}} l = \hat{X}^{K+1} - Y$, then takes a standard backward pass.

For $k = K, \ldots, 1$:

- $\nabla_{Z^k} l = \sigma_k' \left( Z^k \right) \odot \nabla_{\hat{X}^{k+1}} l$

- $\nabla_{\hat{X}^k} l = \left( W_0^k \right)^T \nabla_{Z^k} l$

- $\nabla_{W_0^k} l = \nabla_{Z^k} l \left( \hat{X}^k \right)^T$

where $\sigma'$ is applied element-wise, and $\odot$ is element-wise multiplication.

Now we can already compute $W_b^k = W_0^k - \nabla_{W_0^k} l$. To compute the output tokens, we do another forward pass.

For $k = 1, \ldots, K$:

- $\bar{Z}^k = W_0^k \bar{X}^k - \nabla_{Z^k} l \cdot \mathtt{mask} \left( \left( \hat{X}^k \right)^T \bar{X}^k \right)$

- $\bar{X}^{k+1} = \sigma \left( \bar{Z}^k \right)$

By the end of the forward pass, we have computed $\bar{X}^{K+1}$.

While this forward pass is non-standard, it only contains `matmuls`, `sums`, $\sigma$, and `mask`, therefore is efficient like the standard forward pass.

## A.4. Derivation

To derive the dual form, we show that:

$$\bar{Z}^k = W_0^k \bar{X}^k - \nabla_{Z^k} l \cdot \mathtt{mask} \left( \left( \hat{X}^k \right)^T \bar{X}^k \right)$$

is the same as what would be computed in the primal form. Specifically, we show that each column $\bar{z}_t^k$ of $\bar{Z}^k$ in the second forward pass of the dual equals to $W_t^k \bar{x}_t^k$ in the forward pass of the primal. We invoke a simple fact.

**Fact 1.** *Define matrices* $A = [a_1, \ldots, a_b]$, $Q = [q_1, \ldots, q_b]$, *and* $V = [v_1, \ldots, v_b]$.[3] *Define* $\hat{v}_t = \sum_{s=1}^t a_s^T q_t v_s$, *and* $\hat{V} = [\hat{v}_1, \ldots, \hat{v}_b]$, *then* $\hat{V} = V \cdot mask(A^T Q)$.

Now plug $A = \hat{X}^k$, $Q = \bar{X}^k$, $V = \nabla_{Z^k} l$, and $\hat{V} = W^k \bar{X}^k - \bar{Z}^k$ into the fact above, we have shown the desired equality.

Note that the $\sigma_k$ and $\sigma_k'$ used above can be extended to arbitrary functions that are not necessarily element-wise operations, including normalization layers. This extension can be achieved through, for example, `vjp` (vector-Jacobian product) in standard libraries for automatic differentiation such as JAX and PyTorch. However, the dual form cannot accelerate operations inside $\sigma$ or its `vjp`.

---

[3]Our matrix $A$ would usually be denoted by $K$ in another context. We use $A$ to avoid confusion with the layer number $K$.

# B. Nadaraya-Watson estimator

**Derivation for the Nadaraya-Watson estimator.** Throughout this section, we use $\mathbf{x}$ to denote the input token $x$ as a random variable. Our desired output is the corresponding output token, another random variable $\mathbf{z}$. This is formulated as estimating the conditional expectation of $\mathbf{z}$:

$$\mathbb{E}[\mathbf{z}|\mathbf{x} = x] = \int p(z|x) \, z \, dz = \int \frac{p(x,z)}{p(x)} \, z \, dz.$$

Since the true probability distributions $p(x)$ and $p(x,z)$ are unknown, we replace them with their kernel density estimations. Specifically, the kernel density estimation for $p(x)$ is:

$$\hat{p}(x) = \frac{1}{n} \sum_{i=1}^{n} \kappa(x, x_i),$$

where each $x_i$ is a piece of training data in general. (Recall that for our paper, $x_i$ is specifically training data for the inner loop, *i.e.* a token, which matches our notation in the main text.)

For estimating $p(x,y)$, we use the product kernel:

$$\hat{p}(x,z) = \frac{1}{n} \sum_{i=1}^{n} \kappa(x, x_i) \, \kappa'(z, z_i).$$

At first sight, it seems absurd to factor the joint probability into two seemingly independent kernels. But in this case, $\kappa'$ can actually be any $\kappa'_i$ dependent on $x_i$, since it will be integrated out. So the two kernels do not need to be independent.

Plugging in those estimations, we obtain the Nadaraya-Watson estimator:

$$
\begin{aligned}
\hat{\mathbb{E}}[\mathbf{z}|\mathbf{x} = x] &= \int \frac{\hat{p}(x,z)}{\hat{p}(x)} \, z \, dz \\
&= \frac{1}{\hat{p}(x)} \int \hat{p}(x,z) \, z \, dz \\
&= \frac{1}{\sum_{i=1}^{n} \kappa(x, x_i)} \int \sum_{i=1}^{n} \kappa(x, x_i) \, \kappa'(z, z_i) \, z \, dz \\
&= \frac{1}{\sum_{i=1}^{n} \kappa(x, x_i)} \sum_{i=1}^{n} \kappa(x, x_i) \int \kappa'(z, z_i) \, z \, dz \\
&= \frac{1}{\sum_{i=1}^{n} \kappa(x, x_i)} \sum_{i=1}^{n} \kappa(x, x_i) \, z_i.
\end{aligned}
$$

**Asymmetric kernels.** In modern days, people think of kernels as positive semi-definite, which might not be guaranteed for $\kappa$ unless $\theta_K = \theta_Q$. However, people working on kernels decades ago, around the time when the Nadaraya-Watson estimator was popular, have been very lenient with the choice of kernels, and asymmetric kernels such as our $\kappa$ in Equation 9 have enjoyed a long tradition: When a kernel estimator uses $\theta_K \neq \theta_Q$, it is known as a balloon estimator (Chen, 2017). Papers such as Breiman et al. (Breiman et al., 1977) have even used $\theta_Q$ as a function of $x'$, known as sample-adaptive smoothing.

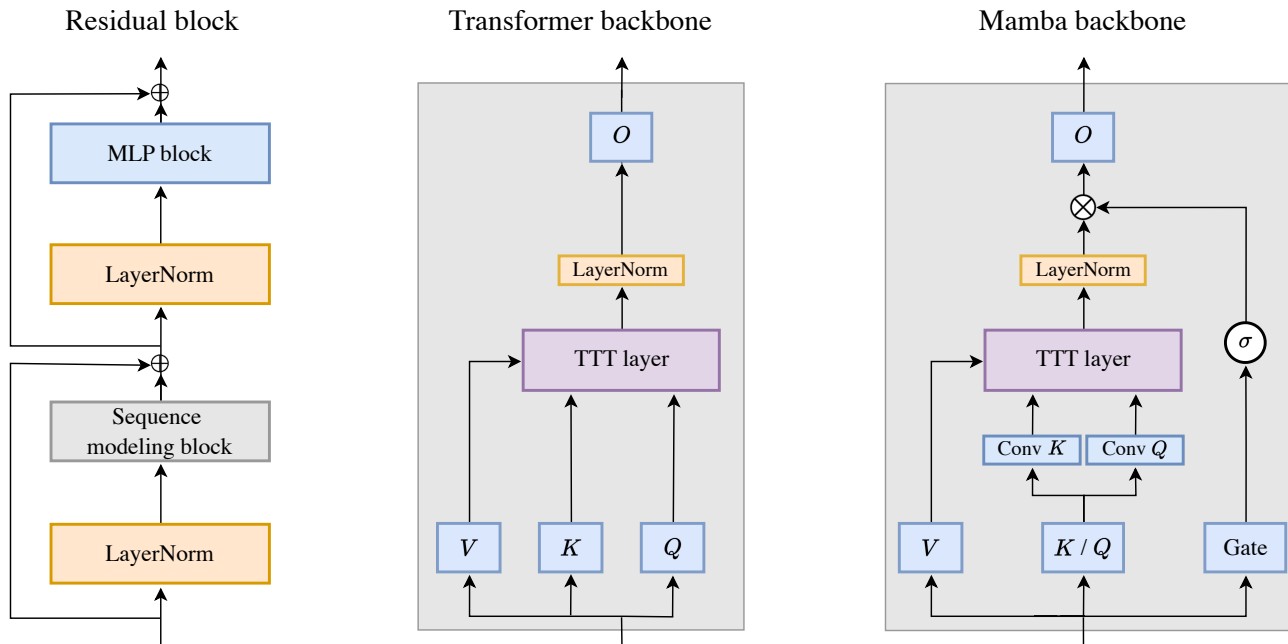

*Figure 9.* **Left**: A residual block, the basic building block for Transformers. The sequence modeling block is instantiated into two variants: the Transformer backbone and Mamba backbone. **Middle**: TTT layer in the Transformer backbone. The LN before $O$ comes from NormFormer (Shleifer et al., 2021). **Right**: TTT layer in the backbone inspired by Mamba (Gu & Dao, 2023) and Griffin (De et al., 2024). Following these two architectures, $\sigma$ here is GELU (Hendrycks & Gimpel, 2016). To accommodate the extra parameters of the gate without changing the embedding dimension, we simply combine $\theta_K$ and $\theta_Q$ into a single projection.

## C. Experiment details

**Architectures.** Our Transformer strictly follows the construction in the Mamba paper, where *Transformer* is called *Transformer++*. Specifically, the Transformer architecture is based on Llama (Touvron et al., 2023), with rotary positional encodings (RoPE) (Su et al., 2023), SwiGLU MLP blocks (Shazeer, 2020), and RMSNorm (Zhang & Sennrich, 2019) instead of LayerNorm. Our Mamba baseline uses the public code provided by the authors. We have verified that our baselines can reproduce the numbers reported in (Gu & Dao, 2023).

**Training configurations.** Our training configurations are in Table 2, which simply reproduces Table 12 in the Mamba paper. All models are trained with a batch size of 0.5M tokens regardless of context length. All of our optimization hyper-parameters follow the "improved recipe" in Appendix E.2 of the Mamba paper, reproduced below:

- AdamW optimizer: $\beta = (0.9, 0.95)$

- Cosine schedule: decay to end learning rate $1e - 5$

- Linear learning rate warmup over 10% of the training steps

- Weight decay: 0.1

- Gradient clipping: 1.0

- No Dropout

- Mixed Precision

For experiments on the Pile, this is the only difference with the recipe in the Mamba paper, which uses two other tokenizers. For experiments on Books, we find that the original angle of the RoPE encoding (Su et al., 2023) $\theta = 10,000$ is sub-optimal

| Params. | Blocks | Embed. dim. | Heads | Train steps | Peak LR | Tokens |
|---------|--------|-------------|-------|-------------|---------|--------|
| 125M    | 12     | 768         | 12    | 4800        | 3e-3    | 2.5B   |
| 350M    | 24     | 1024        | 16    | 13500       | 1.5e-3  | 7B     |
| 760M    | 24     | 1536        | 16    | 29000       | 1.25e-3 | 15B    |
| 1.3B    | 24     | 2048        | 32    | 50000       | 1e-3    | 26B    |

*Table 2.* Training configurations for all experiments. This table reproduces Table 12 in the Mamba paper. The only difference is that the learning rate they use for Mamba and Transformer is $5\times$ the values in their Table 12, and we report the actual values ($5\times$). Note that this table only applies to TTT-Linear, TTT-MLP, and Transformers, as Mamba does not follow the multi-head residual block structure inherited from Transformers.

for our Transformer baseline in long context. Starting at context length 4k, we try $\theta = 500,000$ following the Llama Long paper (Xiong et al., 2023), and use the better perplexity for Transformer (both pretrain and finetune).

**Transformer finetuning.** Finetuning starts a new cosine schedule with the same optimization hyper-parameter as training from scratch, except the peak learning rate. We try three peak learning rates for finetuning: 1e-5, 1e-4, and 1e-3, and select for the best perplexity. We observe that 1e-4 works the best for the 125M models, while 1e-5 works the best for 350M and larger. This observation is reasonable considering that the end learning rate for the Chinchilla recipe is 1e-5.

**Learning rate for TTT.** As mentioned in Subsection 2.7, the inner-loop base learning rate $\eta_{\text{base}}$ is set to 1 for TTT-Linear and 0.1 for TTT-MLP. Our heuristic for setting $\eta_{\text{base}}$ is similar to how people set the outer-loop learning rate for regular training: We tried $\eta_{\text{base}} \in \{0.01, 0.1, 1, 10\}$ and used the largest value that does not cause instabilities. For TTT-MLP, we use linear warmup for $\eta_{\text{base}}$ over 10% of the training steps, similar to regular training. The number of training steps in the inner loop is $T/b$ (assume divisible). For TTT-Linear, we tried linear warmup in the inner loop but did not observe a difference.

**Experiments in Figure 1 (right).** To ensure fairness to Mamba, all methods in these experiments have matched training FLOPs and are trained with the same recipe (last row of Table 2) as Mamba 1.4B. For TTT-Linear and TTT-MLP, matched training FLOPs also imply matched inference FLOPs. Transformer (TF finetune) has $2.8\times$ the inference FLOPs, giving it an advantage as our baseline. To match training FLOPs with Mamba, Transformer has 19 blocks instead of 24. For TTT-Linear and TTT-MLP, their training FLOPs are already close to those of Mamba, so we only need to change the hidden dimension of the MLP blocks from 5504 to 5808 for TTT-Linear and 5248 for TTT-MLP.

**Gradient checkpointing through time.** By default, libraries such as JAX and PyTorch save the intermediate activations during a forward pass so they can be reused during the backward pass. However, for a TTT layer with $W$ as hidden state, this default saves $W_1, \ldots, W_T$, which uses too much memory. With TTT mini-batch and the dual form, we still need to save (assume divisible) $\kappa = T/b$ $W$s at the end of the mini-batches. A standard technique to save memory in this scenario is gradient checkpointing (Chen et al., 2016), which is usually applied through layers, but we apply it through time.

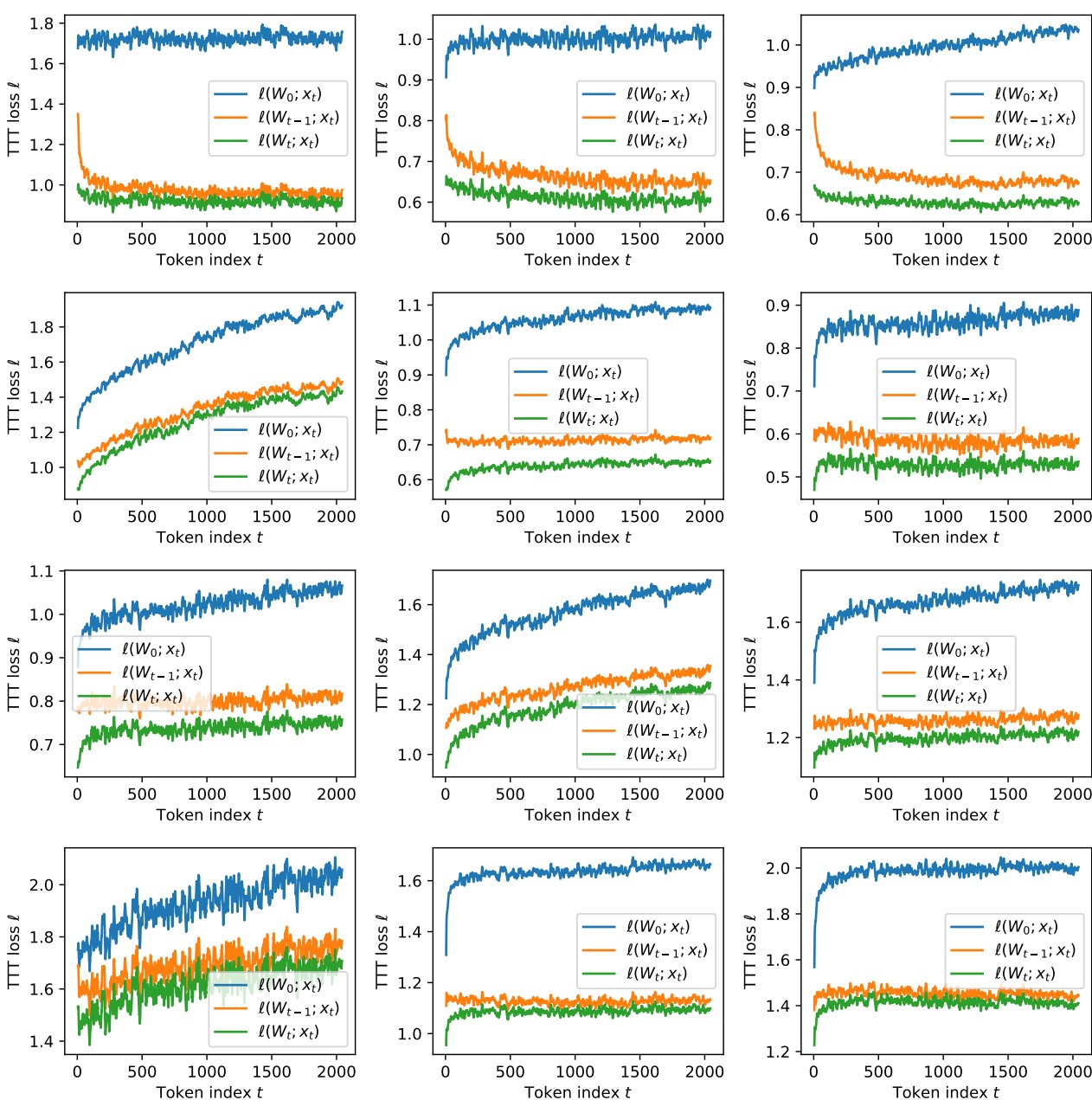

*Figure 10.* The self-supervised TTT loss $\ell$ averaged over all test sequences of the form $x_1, \ldots, x_T$ where $T = 2048$, for all 12 TTT layers in a network with 125M parameters train on the Pile. The same network is also used for $b = 1$ (online GD) in the left panel of Figure 4. For layers in the middle, we observe that $\|x_t\|$ rises steadily, causing all three losses to rise with it. Even for these layers, the gap between $\ell(W_0; x_t)$ and $\ell(W_t; x_t)$ still increases with $t$. For visual clarity, loss values have been averaged over a sliding window of 10 timesteps.

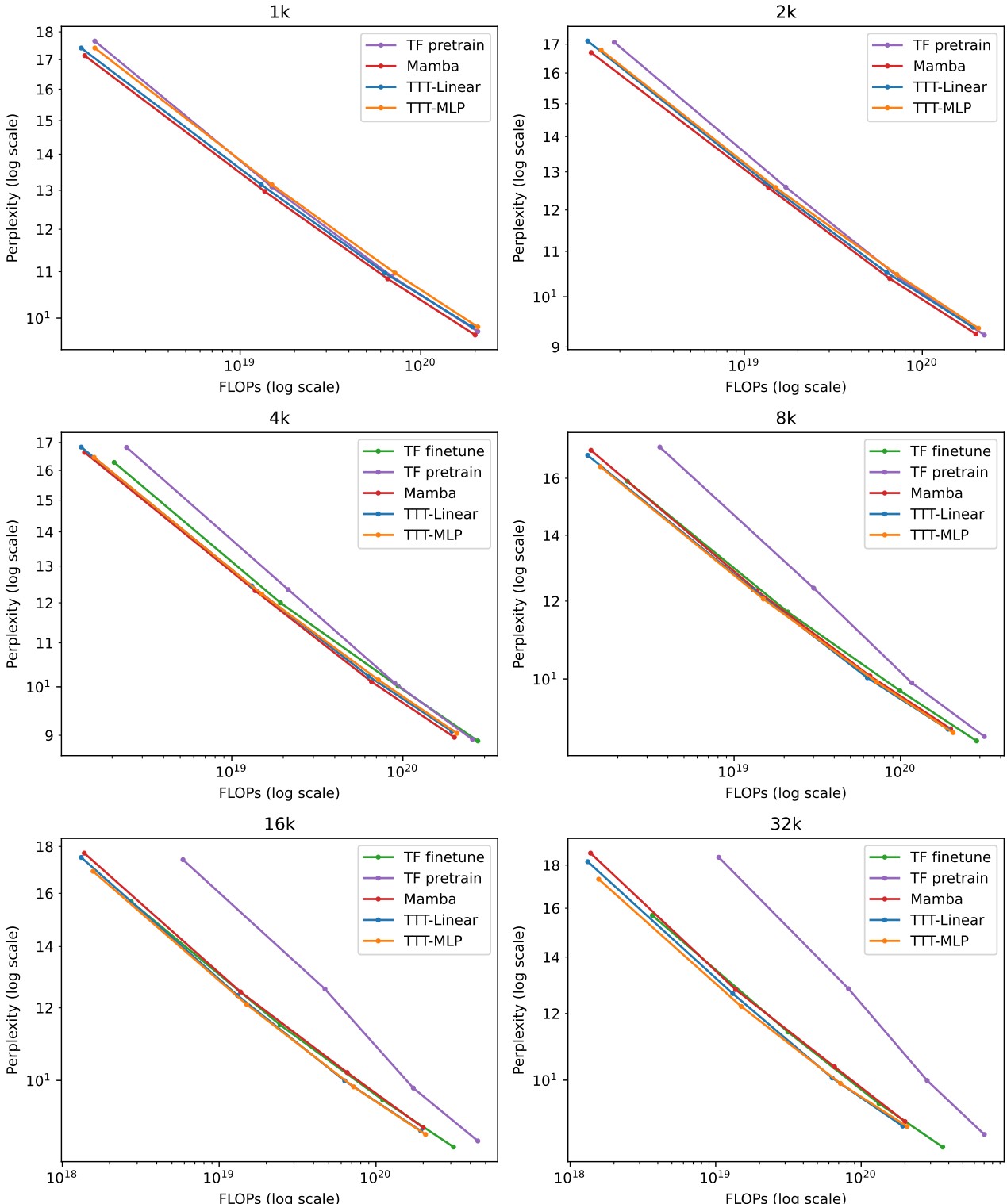

*Figure 11.* Complete results on Books, presented by context lengths. Figure 7 in Subsection 3.2 presents the subset of results for context lengths 2k and 32k.

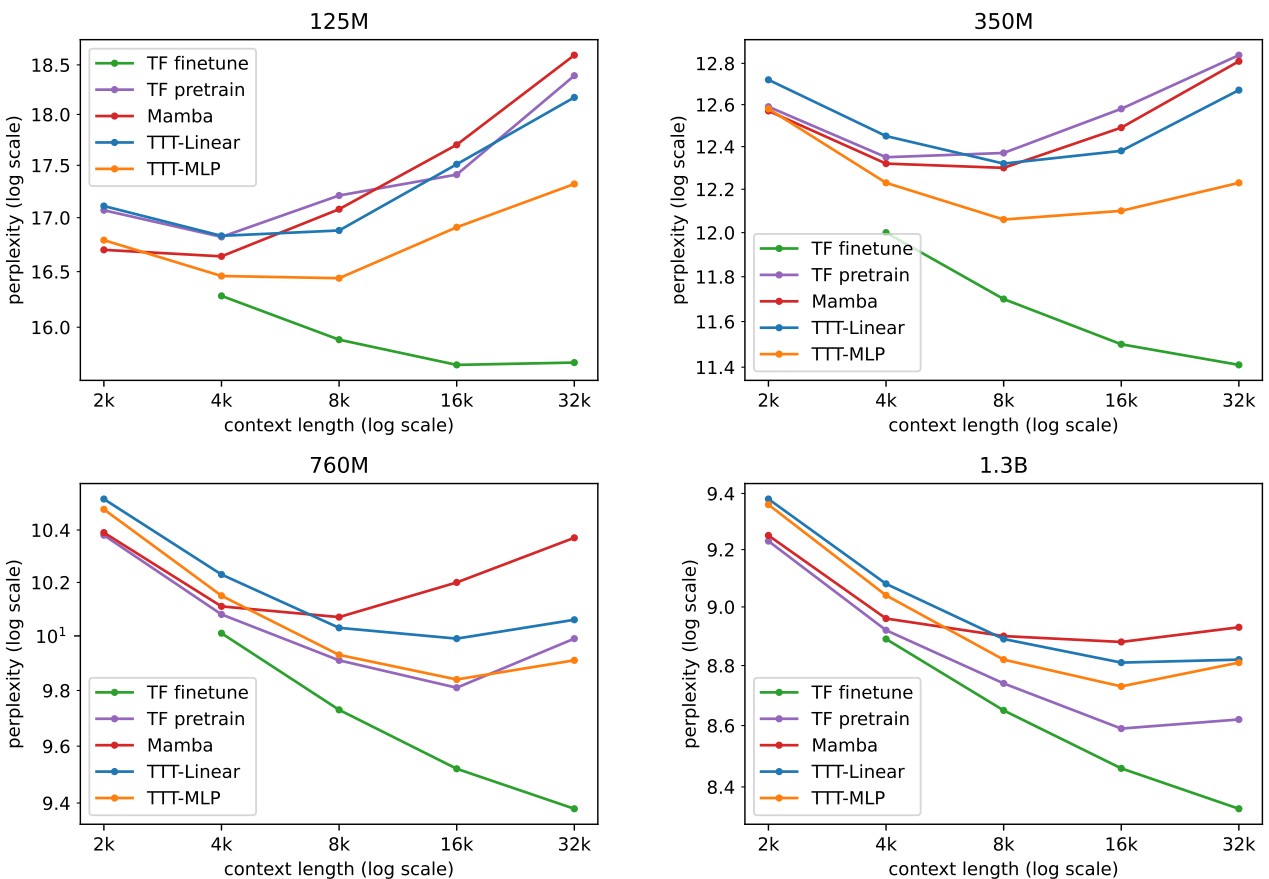

*Figure 12.* An alternative view of our complete results on Books, presented by model sizes, with context length as the x-axis. For all methods trained from scratch, perplexity becomes worse once the context length becomes too large. This trend is not observed with TF finetune, except for one case at the 125M scale. The best context length increases for larger models (trained from scratch).

