# OpenReview forum: "Learning to (Learn at Test Time): RNNs with Expressive Hidden States"
_ICML.cc/2025/Conference — ICML 2025 spotlightposter_

### Official Review · Reviewer_i78k · 2025-02-28

**Overall Recommendation:** 4

**Summary:**

This paper casts the sequence modeling problem as a meta-learning problem at training time. The resulting model is a model which minimizes a loss, i.e. learns at test time.
The authors show that Linear Attention and Attention are special instances in their Learning to Learn at Test Time Framework.
Building on this framework the paper introduces new neural network layers called Test-Time Training (TTT) layers. The paper considers two instantiations of these layers TTT-Linear and TTT-MLP, whose hidden state is a linear model and a two layer MLP, respectively.
In experiments with model sizes ranging from 125M to 1.3B parameters these novel layers match or exceed the performance of Transformer and Mamba baselines.

## Update after Rebuttal:
The authors addressed all my questions and I still recommend acceptance of this paper.

**Claims And Evidence:**

The claims made by this paper are supported by convincing evidence.

**Essential References Not Discussed:**

None.

**Experimental Designs Or Analyses:**

Their experimental design is sound and their ablations in Table 1 shows careful model design.

**Methods And Evaluation Criteria:**

The choice training the models and baselines on the Pile dataset are reasonable. There exist newer, more cleaned open source datasets such as SlimPajama or DCLM Baseline, but these would very likely not change the overall results.

Training on two different context lengths 2k and 8k underlines the fair benchmark setup. For 2k context length the FLOP difference between Transformers (quadratic compute scaling with sequence length) and RNNs (linear scaling with sequence length) is smaller, while for 8k it is larger.

**Other Comments Or Suggestions:**

- It would improve understandability if the final instantiation update rules for the Linear and the MLP variant would be stated somewhere in the paper (including the layernorm and the residual connection).

- You choose the learning rate parameter learnable, so is the learning rate parameter a scalar? Did you think of different parametrizations?
- If one views the learning rate as inter chunk forget gate, can we then explain the improved performance with smaller b in Figure 9 due to a more finegrained (forget-)gating?

**Other Strengths And Weaknesses:**

Strengths:
- Extensive empirical validation of the models.
- Extensive discussion of related work.
- Their proposed model architectures show competitive performance.

Weaknesses:
- The update formulas of their new models (including the dimensions) could be stated explicitly.

**Questions For Authors:**

1) L. 156: Why is the naive layer already efficient in FLOPs? FLOPs not mentioned so far. Do the authors mean the non-quadratic scaling in sequence length compared to transformers?
2) L. 286: What does it mean „all experiments for the TTT layers are performed in one training run“?
Does that mean that there is one model trained with 32k context and then evaluated at different context lengths?
If so which FLOP budget is then plotted on the x-Axis? The training FLOPs?

**Relation To Broader Scientific Literature:**

Even before publication/acceptance at a conference, this paper has inspired several other follow up works.

Moreover, the paper unifies several other linear RNN architectures, such as Mamba, xLSTM (mLSTM), DeltaNet, and many more in a unified framework.

**Theoretical Claims:**

I checked the proofs for Theorem 1 and Theorem 2 for correctness.

---

> ### Author Rebuttal · Authors · 2025-04-01
>
> We thank the reviewer for recognizing the impact of our framework. We also thank the reviewer for the concrete suggestions and questions, which we address below.
>
> ***Explicit update rule for the final instantiations***
>
> Sorry we did not include these formulas in the main text due to space constraints. We chose to defer them to Appendix A (page 14 and 15), which includes a derivation and pseudocode of the update rules in dual form. We will expand Subsection 2.5 in the final version to include at least some of these details.
>
> ***Learning rate parameter***
>
> The learning rate parameter \theta_{lr} is a vector of the same dimension as the embeddings x_t. We have tried making it a scalar and found performance to be slightly worse. We have not tried making it a matrix since that would significantly increase our number of parameters.
>
> ***More fine-grained gating***
>
> We agree with the interpretation that \eta(x) is a gate (more precisely, an input gate). However, we believe that the trend in Figure 9 is not a consequence of \eta(x_t) being input dependent, since we have produced Figure 9 with a fixed \eta and the trend is the same. Intuitively, \eta(x_t) is used for every x_t regardless of the inner-loop mini-batch size b, so larger b will not result in more fine-grained gating.
>
> ***Answering the numbered questions***
>
> 1 - Yes, sorry for the confusion. Perhaps a better phrase would be “efficient in computational complexity.”
>
> 2 - No, it does not mean what the reviewer has guessed. We are sorry for the confusion. Sometimes researchers would have multiple runs of the same experiment but with different random seeds to overcome instability, since some runs would get unlucky. We meant to say that we did not have multiple runs with different random seeds. We will delete this sentence in the final version to avoid possible confusion.

---

> > ### Comment · Reviewer_i78k · 2025-04-02
> >
> > Thanks for the clarifications. Great work!

---

### Official Review · Reviewer_7Qqf · 2025-03-05

**Overall Recommendation:** 4

**Summary:**

The paper proposes the Test-Time-Training (TTT) layer, with the goal to overcome the limitations on the expressive power of modern RNNs. The idea consists in linking the hidden state of an RNN to the parameters of a layer, so that input-driven updates on the hidden state translate into updates of model parameters: by drawing a connection between these updates and gradient descent, the authors effectively manage to setup a training procedure for (a subset of) parameters in the model, which unfolds at test-time. As proven by the authors, the proposed TTT formulation encompasses and generalises the Attention mechanism.
Hardware considerations are employed to speedup the application of the TTT layer, resulting in wall-clock times comparable (or even outperforming) Mamba’s. A detailed analysis of the performance of the layer is reported, including numerous ablations, and scaling behaviour of perplexity of the model when varying context length and size.

**Claims And Evidence:**

For the most part, the evidence proposed is complete enough to allow for a proper assessment of the robustness of the method, and convincing in supporting its validity.
The only claim I would reframe refers to the applicability of TTT to “any neural network as inner model”. While I believe this to be true in principle (as indeed the framework introduced is general enough), in my opinion it fails to address practical considerations which could actually hinder this, and I think it would be fair to better discuss and highlight this limitation in the paper. See also Q1. This is particularly relevant as the generality of the TTT framework is brought up multiple times as one of the main contribution of the paper, setting it aside from previous work (see eg end of Sec4.1 and 4.2).

**Essential References Not Discussed:**

The paper seems to reference the most relevant literature already.

**Experimental Designs Or Analyses:**

To my knowledge, the authors haven’t made their code available, so I can’t check it directly. Regarding the experiment setup, it is quite standard, so I don’t have any particular issue to highlight (see also Methods and Evaluation Criteria, and Q3).

**Methods And Evaluation Criteria:**

The main revolve around comparing perplexity of the proposed model against Mamba and Transformer baselines, after training on the Pile dataset on a fixed FLOPs budget. The sizes of the models and baseline compared are kept similar. The evaluation is in my opinion fair and meaningful. The only improvement I would recommend consists in reporting performances for downstream tasks as well (using, eg, eval_harness framework), to further confirm that perplexity gain indeed do transfer.

**Other Comments Or Suggestions:**

Please change the running title to something more meaningful than “Submission and Formatting Instructions for ICML 2025”

**Other Strengths And Weaknesses:**

The paper is clearly written for the most part, with its core method being described in a concise but effective manner. The idea is interesting and original, although it does build upon other work on modern RNNs and Test-Time Learning. The validity of the method is properly corroborated by the results provided, without over-claiming.

**Questions For Authors:**

Main Questions
1. *Flexibility wrt inner model / loss function* -  While it’s true your model formulation is flexible enough to accommodate for generic inner model and loss functions in theory, in practice wall-clock time remains a concern. As your layer aims to replace Attention or the Mamba mixer, it is reasonable to expect its application to require a comparable computational time. I really appreciated in this sense the comparison in Fig6, but the results therein already highlight how much just introducing the simplest nonlinearity (an MLP) impacts wall-clock time, due to the added complexity in both (inner) forward and backward computations. More complicated loss functions, or layer whose gradient is not readily computable, or that involve additional matrix-matrix-multiplications, are all factors which will further exacerbate this. If you agree with this, it should be properly reported as a limitation of the method; if not, please elaborate.


Minor

2. *On using a different loss* - Given the improvement that mini-batching provides to Linear Attention (ie, TTT-linear with $b=T$, as per Thm1), I was looking forward to seeing a similar application to self-attention, (leveraging Thm2). Also in light of my comment in Q1, I understand why you decided to stick with the simpler inner-loss definition, but it would be interesting to try (especially as I agree with your remark that the choice of inner loss would likely impact the performance of the method), and was wondering whether you had already started experimenting on it.

3. *Fig6* - Is Fig6 reporting average time per token at *training* or *inference*? The reason I’m asking this, is because when processing input data at training, I was expecting to see a (slight) increase of wall-clock time of TTT with context length. This is in light of the fact that, while computing gradients updates *within* a mini-batch is parallelisable (as you show and take advantage of), the TTT gradient updates *along* mini-batches is a sequential procedure (as you need to compute the various updated $W_{0}, W_{b}, W_{2b},…$ in order). Of course this doesn’t hold for generation, since no form of sequence length parallelisation is applied anyway. If Fig6 actually reports average inference time at generation, as I suspect, could you correct its title and also include average time for a forward pass during training (just to get an idea of how much more/less time consuming it is to train your model vs Mamba or Transformer). If not, what am I missing?

4. *Fig7* -  Why would $l(W_0,x_t)$ increase over $t$ in Fig7? What causes the rise in $||x_t||$? Which Positional Encoding do you use?

**Relation To Broader Scientific Literature:**

The project expands upon two main branches of work: modern RNNs, and Test-Time-Adaptation. Regarding modern RNNs, for its mini-batching procedure the paper reuses a similar chunk-splitting concept as in GLA (Yang et al, 23); however, in GLA this splitting is mostly a reorganisation of the operations to improve hardware efficiency, while in TTT it’s used to introduce an actual sequential update. Most important are the similarities to DeltaNet (Yang et al, 24): indeed, for specific choices of its inner model and loss function, TTT is exactly equivalent to DeltaNet. The authors however expand on this framework (with some caveats, see Q1). Regarding Test-Time-Adaptation, the method directly inherits the core idea of training a net during inference. In either case, in my opinion the discussion in Sec4 covers these connections reasonably well and fairly.

**Theoretical Claims:**

The main theoretical claims aim at drawing correspondences between the proposed method and linear / softmax attention, when specific choices on the TTT loss function and inner model are made. They also show the equivalence of primal and dual formulation of their method. The proofs mostly reduce to algebraic manipulations: I’ve checked their derivations and they appear correct.

---

> ### Author Rebuttal · Authors · 2025-04-01
>
> We thank the reviewer for the insightful questions, which we answer below.
>
> ***1 - Flexibility w.r.t. inner model***
>
> We agree. We will add the discussion below to the final version if the paper is accepted (ICML does not allow submitting a revision):
>
> In principle, any differentiable model and loss function can be instantiated as the inner-loop model and loss in our framework, but the practical effectiveness of such instantiations can be limited by their requirements in wall-clock time. While TTT-MLP is an effective instantiation in FLOPs, the additional complexity of the MLP structure makes the increase in wall-clock time much larger than the increase in FLOPs, as shown in Figure 6. It remains to be seen whether our framework can produce instantiations that overcome this limitation.
>
> ***2 - On using a different loss***
>
> Yes, a different inner loss among the most important things to try next. For the past few months, we have been experimenting with using next-token prediction as the inner loss and a Transformer as the inner model. Our initial results so far have been quite promising.
>
> ***3 - Figure 6***
>
> Figure 6 is for training (forward and backward). Timing for inference is in Figure 13 in the Appendix, which has a separate plot for prefill (forward) and decode (generation). We appreciate that the reviewer is clear about the difference between training, prefill and decode.
>
> The reviewer said: *"I was expecting to see a (slight) increase of wall-clock time of TTT with context length."*
>
> A linear-complexity method that is completely parallelizable across the entire sequence should see a decrease in latency as context length increases, while one that is completely sequential should have constant latency. This is because latency = processing time / tokens processed. As tokens processed increases, the processing time for a parallelizable method increases at a much slower rate, while that for a sequential method increases at the same rate. In Figure 6, none of the methods exhibit significant decrease in latency, showing that given the compute resources in Figure 6, they are all far from parallelizable.
>
> ***4 - Figure 7***
>
> We appreciate the reviewer’s attention to details. We use RoPE, but we do not think the increase in l(x_t;  W_0) has to do with the position embedding. As the reviewer already knows, the increase in l(x_t;  W_0) is a consequence of the increase in \|x_t\|, whose reason is unclear to us. We have measured \|x_t\| in the Transformer baseline and observed a similar increase, so we know that it is not caused by using TTT layers.
>
> We will also change the running title, thanks for catching that. Regarding reproducibility, we had to delete the link to our online code repo in the manuscript for anonymity. Many people have been able to reproduce our results using the code online. We will add a link to the final version if the paper is accepted.

---

### Official Review · Reviewer_L8fP · 2025-03-16

**Overall Recommendation:** 5

**Summary:**

This paper introduces Test-Time Training (TTT) layers, a 'clever' way to handle long sequences without the heavy cost of Transformers. The key idea is to treat the hidden state as a learning model that updates itself during inference, allowing it to capture complex patterns over long contexts. The authors propose two versions, TTT-Linear and TTT-MLP, which outperform existing models like Mamba but also runs faster on modern hardware.

**Claims And Evidence:**

Yes

**Essential References Not Discussed:**

I think (Irie, Kazuki, et al. "A modern self-referential weight matrix that learns to modify itself." ICML, 2022.), which also combines ideas of FWP, RNNs, meta-learning through recursive self-improvement,  is extremely relevant here and should be discussed in this paper, with key differences highlighted.

**Experimental Designs Or Analyses:**

The experimental design is limited but sufficient for a proof of concept. I think they could have included another task beside language modeling, e.g., time series.

**Methods And Evaluation Criteria:**

Yes

**Other Comments Or Suggestions:**

Nan

**Other Strengths And Weaknesses:**

Paper is very well-written and structured

Weaknesses:
- The main weakness of this work is the limited experimental setting and comparisons. Including results with relevant baselines, e.g., DeltaNet and Mamba2 would make the claims stronger and help us see how good these models are compared to these baselines.

- The authors don't share any code for reproducing their results. So hard to judge the reproducibility.

**Questions For Authors:**

1- Why not Including competitive results for Mamba2 and DeltaNet?

2- Can you comment on the similarities and differences compared to  (Irie, Kazuki, et al. "A modern self-referential weight matrix that learns to modify itself." ICML, 2022.)?

3- Why did use mamba as backbone for TTTLinear and TTT-MLP? why not recent SSM models, such as mamba2? How does the model perform with such backbones? I understand that (mamba-->mamba2) is an orthogonal improvement to the contribution proposed here but would it be better to include the results with the top performant backbones to assess the full potential of TTT?

4- the claim in first paragraph in Section 4.1. '' The hidden state in these models is a vector, similar to in LSTMs. For TTT-Linear or TTT-MLP, the hidden state is a matrix or two matrices, therefore larger'' is not valid for all SSMs. In fact, recent SSMs, such as (FACTS: A Factored State-Space Framework For World Modelling) the hidden state is already a matrix.

5- When the hidden state  of the backbone is already a Matrix (e.g., FACTS), does using TTT still improve the expressiveness of the model or this gain is reduced? would it affect the computational aspects of TTT?

6- Is it possible to share code to reproduce some parts of your results (e.g., on the Pile) otherwise, it is hard to assess the reproducibility of the approach's results.

7- One of the major issues of SSMs models and Mamba in particular (https://arxiv.org/pdf/2501.00658) is the over-smoothing and sensitivity to noise, especially at the end of the sequence (See table 1 in https://arxiv.org/pdf/2501.00658). I wonder how does test-time training affect this behavior? Can the authors share their thoughts on it. Because if it is the case, this can mitigate one of the current major bottlenecks of SSMs.

**Relation To Broader Scientific Literature:**

This paper builds on prior work in sequence modeling,(especially  Schlag et al., 2021 and Yang et al. 2024), while addressing their limitations in long-context settings. By leveraging self-supervised learning and meta-learning techniques, the proposed Test-Time Training (TTT) framework introduces adaptive hidden states which might be the way to improve both efficiency and expressiveness of SSMs

**Theoretical Claims:**

Yes

---

> ### Author Rebuttal · Authors · 2025-04-01
>
> We thank the reviewer for appreciating our paper as a sufficient proof-of-concept. We also thank the reviewer for the insightful questions, which we answer below.
>
> ***1 - Comparison with Mamba 2***
>
> Mamba 2 130M with 2k context trained with Chinchilla scaling law on the Pile has perplexity 10.77, which is significantly lower than our number of 11.09 reported in Table 1 of the manuscript. However, 11.09 was based on the older Mamba backbone. During the rebuttal period, we experimented with putting TTT-Linear into the Mamba 2 backbone. This model has perplexity 10.65, which is slightly better than Mamba 2.
>
> ***2 - Discussion on Irie et al. 2022***
>
> We thank the reviewer for bringing up this very relevant work. We will add the discussion below to the final version if the paper is accepted (ICML does not allow submitting a revision):
>
> Irie et al. 2022 is similar to DeltaNet, except that the key, query and value vectors are also produced by the hidden state matrix. In the terminology of TTT, all outer-loop parameters are the initializations of inner-loop parameters. This approach naturally situates Irie et al. 2022 in the same setting as MAML (Finn et al. 2016), and their experiments are both on few-shot learning.
>
> ***3 - Results with Mamba 2 backbone***
>
> Answer for your question 1 in this rebuttal switches to Mamba 2 backbone. As the reviewer expected, the improvement in backbone is indeed orthogonal to the improvement we had with TTT layers. Mamba 2 had not been released when we conducted most of the experiments in May, 2024.
>
> ***4 - Incorrect claim about related work***
> Indeed, we have missed some of the recent developments in SSMs, making our claim incorrect. We appreciate the reviewer for carefully reading our text and pointing out this error. We will make sure to fix this in the final version if the paper is accepted.
>
> ***5 - Improvements on top of linear hidden states***
>
> The hidden states of linear attention and Mamba 2 are already matrices. Compared to linear attention, TTT-Linear brought significant improvements because of the added nonlinear operations. Compared to Mamba 2, TTT-Linear still brings a small improvement.
>
> ***6 - Sharing code for reproducibility***
>
> Sorry we forgot to upload a zip file for ICML submission. We had to delete the link to our online code repo in the manuscript for anonymity. Many people have been able to reproduce our results using the code online. We will add a link to the final version if the paper is accepted.
>
> ***7 - Tradeoff between sensitivity to noise vs. over-smoothing***
>
> We thank the reviewer for pointing out this interesting paper. More expressive hidden states will improve both ends of this tradeoff. As discussed in the paper, the key that controls this tradeoff is recency bias: too much recency bias results in sensitivity to noise, but too little results in over-smoothing. RNN layers with expressive hidden states will be able to remember useful information (therefore less over-smoothing) without using aggressive recency bias (therefore less sensitivity to noise), simply because it has a larger capacity for information. Take self-attention as an example. It does not need any recency bias because it simply stores all key-value pairs in a cache. We believe that very expressive inner models for TTT can achieve similar effects.

---

### Official Review · Reviewer_SWT9 · 2025-03-17

**Overall Recommendation:** 3

**Summary:**

This paper presents a novel approach to sequence modeling by enhancing the hidden states of RNNs through a general method called Test-Time Training (TTT). The key idea is to frame the online updating of hidden states as a self-supervised learning process, using the update loss $\ell: W_t = W_{t−1} − \eta \nabla \ell(W_{t−1}; x_t)$, where $W_t$ is the hidden states learnt at each step, reminiscent of fast weights in RNN literature, and $\nabla$ denotes gradients from the chosen objective function. This framework unifies several previous approaches, including naive RNNs, linear attention, and even self-attention, under a common paradigm.

One main challenge of TTT is the difficulty of parallelization, as the proposed objective may introduce non-linearities and dependencies on previous steps. To address this, the authors propose a batch update approach, allowing $W_t$  to depend on the previous N steps, or in extreme cases, on $W_0$. Interestingly, the authors demonstrate that under certain conditions, TTT becomes equivalent to vanilla linear attention.

Building on this foundation, the authors introduce two variants: TTT-linear and TTT-MLP, both incorporating layer normalization and mini-batch updating. The experimental results on approximately 1B parameter models show promising performance in both short and long-context scenarios, demonstrating the potential of TTT for sequence modeling tasks.

**Claims And Evidence:**

No.

I really appreciate the TTT framework proposed in this paper. However, the experiments conducted are somewhat limited, making the validation of TTT less comprehensive and persuasive.

I strongly recommend the authors include additional common benchmarks found in related works like Mamba or GLA, such as hellaswag, lambada_openai, and winogrande, beyond just measuring perplexity on text length.

Furthermore, more detailed comparisons with other relevant models like DeltaNet (which is heavily discussed in the paper) and Mamaba2 would significantly strengthen the evaluation.

**Essential References Not Discussed:**

Recent concurrent works should be included and discussed:
1. Unlocking State-Tracking in Linear RNNs Through Negative Eigenvalues
2. gated delta Networks: Improving Mamba2 with Delta Rule
3. Longhorn: State Space Models are Amortized Online Learners

**Experimental Designs Or Analyses:**

Yes

**Methods And Evaluation Criteria:**

Yes

**Other Comments Or Suggestions:**

See below.

**Other Strengths And Weaknesses:**

1. The experimental section of this paper is insufficiently comprehensive, making the results less convincing.
2. The detailed implementations of TTT-MLP and TTT-Linear are not well-explained, which is quite different from the naive cases in Sec 2.5.

**Questions For Authors:**

I'm having trouble understanding how TTT-Linear and TTT-MLP work.
Given that both methods include Layernorm, which might create obstacles for parallelism and make it difficult to accelerate using tensor cores.

Could the authors explain how the dual forms introduced in Section 2.5 can be implemented for TTT-Linear and TTT-MLP?

**Relation To Broader Scientific Literature:**

I can see potential benefits in using TTT to develop more efficient memory management strategies for RNNs, guided by the principles of TTT.

**Theoretical Claims:**

Yes

---

> ### Author Rebuttal · Authors · 2025-04-01
>
> We thank the reviewer for recognizing our framework’s potential to guide the development of RNNs with better memory. We also thank the reviewer for the concrete suggestions and questions, which we address below.
>
> ***Benchmark on downstream tasks beyond perplexity***
>
> The focus of our paper is on long context. However, all the downstream tasks in Mamba (Gu and Dao, 2024), as the reviewer suggested, have sequence length <1000. In fact, most of them only need ~100 tokens. There also exist downstream tasks with long context, such as book summarization and solving software issues in large repositories. However, these tasks require post-training for instruction following capability, and pre-training frontier models with trillions of tokens. Unfortunately, the cost of this evaluation exceeds our budget.
>
> ***Comparison with Mamba 2***
>
> Mamba 2 130M with 2k context trained with Chinchilla scaling law on the Pile has perplexity 10.77, which is significantly lower than our number of 11.09 reported in Table 1 of the manuscript. However, 11.09 was based on the older Mamba backbone. During the rebuttal period, we experimented with putting TTT-Linear into the Mamba 2 backbone. This model has perplexity 10.65, which is slightly better than Mamba 2.
>
> ***Concurrent work***
>
> We appreciate these references highlighted by the reviewer. We will add the discussion below to the final version if the paper is accepted (ICML does not allow submitting a revision):
> Concurrent work such as [1] [2] [3] has further improved the update rule for RNNs with matrix hidden states. Given that TTT-Linear is already similar to DeltaNet and linear attention, it is also similar to these more recent variants. As discussed in the opening of the paper, our core contribution is a practical framework to use any neural network as the hidden state of an RNN. This contribution is still salient in comparison with concurrent work.
>
> ***Would LayerNorm create obstacles for parallelism? How to implement the dual form for TTT-Linear and TTT-MLP?***
>
> Sorry we did not include these details in the main text due to space constraints. We chose to defer them to Appendix A (page 14 and 15), which includes a derivation and pseudocode of the dual form for neural networks of arbitrary depth with arbitrary nonlinear operations. We will expand Subsection 2.5 in the final version to include at least some of these details.
>
> Following the explanation in Appendix A, it should be clear that LayerNorm (or any other nonlinear operations with few FLOPs) would not create an obstacle for parallelizing the rest of the hidden state. While LayerNorm runs on CUDA Cores, compute-intensive operations, specifically matrix multiplications, will still run on Tensor Cores  As an intuitive analogy, regular neural networks, such as Transformers, also have many nonlinear operations, but all of their linear operations can still be parallelized.
>
> Does the reviewer feel that we have adequately addressed their concerns? Please let us know if we can provide more information.

---

### Decision · Program_Chairs · 2025-05-01

**Decision:**

Accept (spotlight poster)

**Comment:**

**Overall review**
There is consensus towards accepting this work. Reviewers coincide this is a well-written paper, with good results and solid theory. They all lean towards acceptance. I also believe this work is interesting as it generalizes transformers and RNNs and can spark new research in the topic.

**Summary**
The paper introduces a novel sequence modeling approach . The key idea is to treat the hidden state of a recurrent neural network (RNN) becomes the weights of a learnable model that updates itself during inference using a meta-learned self-supervised task. This allows the model to capture complex patterns over long contexts more efficiently than traditional Transformers.

**Reviews**

Reviewer SWT9:

Strengths:
Appreciates the Test-Time Training (TTT) framework.
Sees potential benefits for efficient memory management in RNNs.

Weaknesses:
Limited experimental validation. (Partially addressed: Authors provided new results comparing to Mamba2)
Missing comparisons with relevant models (DeltaNet, Mamba2). (Partially addressed: Authors provided new results comparing to Mamba2)
Unclear implementation details of TTT-MLP and TTT-Linear. (Addressed: Authors mentioned that details were in the appendix and would expand the section in the final version.)


Reviewer L8fP:

Strengths:
Well-written and structured paper.
TTT offers a "clever" way to handle long sequences efficiently.
Potential to improve efficiency and expressiveness of SSMs.

Weaknesses:
Limited experimental setting (only language modeling). (Not addressed)
No code provided for reproducibility. (Addressed: Authors said they would add a link to the online code repo in the final version.)
Missing comparisons with relevant baselines (DeltaNet, Mamba2). (Partially addressed: Authors provided new results comparing to Mamba2)


Reviewer 7Qqf:

Strengths:
Complete evidence to assess the robustness of the method.
Interesting and original idea.
Clear and concise writing.
Properly corroborated results without over-claiming.

Weaknesses:
Applicability of TTT to "any neural network as inner model" needs more practical consideration. (Addressed: Authors said they would add a discussion about this limitation in the final version.)
Downstream task performance not reported. (Not addressed)


Reviewer i78k:

Strengths:
Extensive empirical validation.
Extensive discussion of related work.
Competitive performance.
Unifies several linear RNN architectures in a framework.

Weaknesses:
Explicit update formulas (including dimensions) for new models are missing. (Addressed: Authors mentioned that details were in the appendix and would expand the section in the final version.)